# A Histological Evaluation of the Efficiency of Using Periprosthetic Autologous Fat to Prevent Capsular Contracture Compared to Other Known Methods—An Experimental Study

**DOI:** 10.3390/diagnostics14060661

**Published:** 2024-03-21

**Authors:** Mihaela Pertea, Nadia Aladari, Oxana Madalina Grosu, Stefana Luca, Raluca Tatar, Andrei-Nicolae Avadanei, Madalina Palaghia, Ana Maria Trofin, Sorinel Lunca, Nina Filip

**Affiliations:** 1Department Plastic Surgery and Reconstructive, Faculty of Medicine, “Grigore T. Popa” University of Medicine and Pharmacy, 700115 Iasi, Romania; mihaela.pertea@umfiasi.ro; 2Department of Plastic Surgery and Reconstructive Microsurgery, “Sf. Spiridon” Emergency County Hospital, 700111 Iasi, Romania; naladari@yahoo.com; 3Faculty of Medicine, “Carol Davila”University of Medicine and Pharmacy, 020021 Bucharest, Romania; raluca.tatar@umfcd.ro; 4Department of Plastic Reconstructive Surgery and Burns, “Grigore Alexandrescu” Clinical Emergency Hospital for Children, 011743 Bucharest, Romania; 5Department of Vascular Surgery, “Sf Spiridon” Emergency County Hospital, 700111 Iasi, Romania; andrei.n.avadanei@gmail.com; 6Department of Surgery, Faculty of Medicine, “Grigore T. Popa” University of Medicine and Pharmacy, 700115 Iasi, Romania; madalina.palaghia2@umfiasi.ro (M.P.); ana-maria.trofin@umfiasi.ro (A.M.T.); 7Surgery Clinic, “Sf Spiridon” Emergency County Hospital, 700111 Iasi, Romania; 8Second Oncological Clinic, Regional Institute of Oncology, 700483 Iasi, Romania; sorinel.lunca@umfiasi.ro; 9Department of Morpho-Functional Sciences (II), Faculty of Medicine, “Grigore T. Popa” University of Medicine and Pharmacy, 700020 Iasi, Romania; nina.zamosteanu@umfiasi.ro

**Keywords:** silicone implant, capsular contracture, autologous fat, histological results, rats

## Abstract

Background: Capsular contracture remains a common complication in silicone breast implantation. The etiology, formation mechanisms, predisposing and favoring factors are still subjects of research. This study aims to demonstrate the effectiveness of using autologous fat introduced periprosthetically in preventing capsular contracture compared to other known methods: antibiotics and corticosteroids. Methods: A cohort of 80 Wistar rats was included in the study, divided into four subgroups. All subjects received a silicone implant, implanted in a pocket created along the abdominal midline. The first subgroup served as the control group, with subjects having the implant placed without any treatment. For the second and third subgroups, the implants were treated with an antibiotic solution and intramuscular injections of dexamethasone, respectively. The subjects in the last subgroup received centrifuged autologous fat introduced periprosthetically. Results: The subgroup with autologous fat exhibited a significantly smaller capsule thickness, which was poorly represented, with a smooth surface. The use of autologous fat for treating silicone prosthesis was linked with the lack of acute inflammation around the prosthetic site. Conclusions: Autologous fat helps to minimize the “non-self” reaction, which results in the development of a periprosthetic capsule consisting of mature collagen fibers interspersed with adipocytes.

## 1. Introduction

Breast augmentation or reconstruction using silicone-based implants ranks among the most frequently performed procedures in both reconstructive and aesthetic surgery [1,2,3]. In recent years, one of the most prevalent complications observed is capsular contracture, which can be distressing for patients both in terms of aesthetics and functionality [4,5]. The development of the periprosthetic capsule is described as a natural part of the healing process, with some studies indicating its usefulness in anchoring the implant in the receptor site [5,6,7]. Nevertheless, when this capsule thickens and becomes contractile, thereby distorting the dimensions of the implants, the condition is referred to as capsular contracture [8]. The occurrence of capsular contracture seems to range between 8% and 15%, and it was clinically classified by Baker in 1975 into four grades [6,9,10,11]. The exact cause of capsular contracture is not entirely understood and is likely multifactorial. Numerous hypotheses have been proposed, including the theory of an inflammatory process triggered by subclinical infection, particularly in asymptomatic patients, which results in the formation of hypertrophic scars. The theory of an immunologic response has surfaced from studies that have identified heightened fibrosis indices and other humoral factors directly linked to the extent of contracture [12,13,14,15]. Additional theories outlined in the literature include those related to hematoma formation, a foreign body reaction, and theories involving the activation of myofibroblasts [16]. Capsular contracture appears to be directly impacted by factors such as the implant’s content (silicon or a saline solution), implant’s placement (submuscular or subglandular), implant’s surface texture (smooth or textured) and bacterial infection. Moreover, patients undergoing more intricate procedures, like breast reconstruction, tend to exhibit higher rates of capsular contracture compared to those undergoing simpler augmentation [17,18,19].

Methods of prevention, apart from those related to implant texture and content, include the techniques used for implant placement, the surgical approach, the placement site and method, the use of drainage, and the wound closure method [17,20,21]. Adams et al. consider antibiotic prophylaxis to be extremely important, proposing the intraoperative washing of the implant pocket with a triple combination of antibiotics, a procedure proven to be highly effective in preventing capsular contracture [22,23]. The use of glucocorticoids in the prevention (or treatment) of capsular contracture has significant potential; however, the doses required for therapeutic efficacy result in significant systemic side effects [24,25]. Moreira et al. tested the effect of locally administering a single dose of liposomal prednisolone phosphate (PPL) on capsule formation and observed a significant decrease in capsule thickness and collagen density, as well as a significant reduction in the number of myofibroblasts [25]. Over time, other treatment techniques for capsular contracture have been brought to attention, specifically addressing its occurrence, such as an autologous fat transfer. This can be used as a lipofilling by injecting it in the vicinity of the capsule with the aim of reducing the formation of capsular contracture. Alternatively, it can be employed as a fat graft, used simultaneously with implant placement. Positive results have been reported, especially in the case of breast reconstruction, with autologous fat transfer being used alongside silicone implantation. The literature’s reports indicate that an autologous fat transfer improves the vascularization around the implant [26].

Another method mentioned for preventing capsular contracture is wrapping the implant in a layer of an acellular dermal matrix, which reduces the myofibroblast layer formed around the implant, resulting in decreased myofibroblast proliferation and inflammatory processes, thus reducing the risk of long-term capsular contracture [27,28,29].

The current study compares the histological results obtained from the analysis of periprosthetic capsules using various preventive methods: the local administration of an antibiotic (rifampicin), dexamethasone administration, and autologous periprosthetic fat use, compared to a control group in which none of the reported preventive methods in the literature were used. The introduction of periprosthetic autologous fat will lead to the formation of a delicate, uniform, loose, and highly elastic conjunctive capsule.

## 2. Materials and Methods

The research was carried out on a cohort of eighty adult Wistar rats, weighing between 360 and 420 g. These rats were divided into four study groups and kept under consistent temperature and humidity conditions. They were provided standard rat feed and water as per laboratory protocols and subjected to 12 h light–dark cycles. Before the surgery, the rats were placed in separate cages labeled according to the treatment they would receive, which aided in the monitoring and data collection throughout the study.

The study received ethical approval from the “Grigore T. Popa” University of Medicine and Pharmacy in Iași. Wistar rats were housed, operated on, monitored, and cared for in the research laboratory of the Faculty of Pharmacy within the “Grigore T. Popa” University of Medicine and Pharmacy in Iași. The animals were allowed to adapt to the new environment for one week, after which they underwent surgical intervention for the introduction of the silicone implant, were monitored for 6 weeks, followed by a second surgical intervention for the extraction of the silicone implant and the corresponding formed capsule. Prior to surgery, the subjects were sedated with ether vapors and subsequently anesthetized via an intramuscular injection of Ketamine (100 mg/mL) (VetViva Richter GmbH Richter Pharma AG, Wels, Austria) at the dosage of 0.3 mL/kg, along with Xylazine (2%) (Bioveta a.s., Ivanovice na Hané, Czech Republic) at a dosage of 0.2 mL/kg. The anticipated onset of anesthesia was approximately 3 min following administration. The implant utilized was a microtextured silicone implant tailored to fit the dimensions of the subjects, each containing 2 cc (with a 2 cm diameter) in accordance with an institutionally approved protocol for researching prosthetic tolerance (Figure 1).

Each study group included 20 rats. The first group served as the control, with subjects receiving a mammary implant without any processing, and no other treatment was administered to the animals. In the second group, the silicone implant was immersed in a solution of rifampicin (60 mg of rifampicin in 25 mL of saline) before being inserted into the pocket created along the abdominal midline toward the mammary line. The subjects in the third group received injectable treatment with intramuscular dexamethasone at a dosage of 0.2 mL per day for 10 days following the introduction of the silicon implant. In the fourth group, autologous centrifuged periprosthetic fat collected from the inguinal fold of each subject was introduced.

To acquire the autologous fat emulsion, a high-precision clinical centrifuge was employed, operating at speeds ranging between 100 and 5000 rotations per minute (rpm), with a maximum force of 3074 g. The centrifuge featured a fixed rotor angle of 45 degrees and a timer ranging from 0 to 99 min, accommodating up to 8 tubes with volumes ranging between 9 and 15 mL. The obtained 1.5 mL emulsion was introduced peri-implant using a syringe. At the conclusion of the study, all rats utilized were euthanized following the euthanasia protocol outlined by the American Veterinary Medical Association (AVMA), using ketamine (100 mg/mL) as an anesthetic.

The collection of anatomopathological samples was performed in a special necropsy room. The analyzed parameters included periprosthetic capsule formation, the capsule thickness, pericapsular vascularization, the lymphocytic infiltrate at the capsule level, and the inflammatory cellularity. After collection, peripheral tissue fragments of the implant were fixed in 10% buffered formaldehyde for 48 h, then processed using the paraffin embedding method (Leica TP 1020 processor—Nussloch, Germany) and sectioned (SLEE microtome, Nieder-Olm, Germany) at a thickness of 5 μm. Sections were stained (bicromic) with hematoxylin-eosin and examined with a photonic microscope (Leica DM 1000, Wetzlar, Germany), using a digital histological camera (Leica 5 MPx, full HD, Wetzlar, Germany) and LAS software, version 2016, for image capture. The collected periprosthetic capsule samples were fixed in 10% buffered formalin. After 48 h, they were embedded in paraffin.

Sections were made at a size of 2–5 microns for evaluating the capsular architecture. Subsequently, the staining with hematoxylin-eosin (HE) was performed to assess the neoformed periprosthetic connective tissue. Several parameters were evaluated for the qualitative and quantitative assessment of periprosthetic tissue: the degree of maturation (fibroblast/collagen, fiber/collagen, fiber appearance, and density), the conjunctive capsule thickness, the presence of inflammatory processes, neovascularization, and the appearance of the contact surface with the breast prosthesis. Periprosthetic inflammation in the four groups was also analyzed.

### Statistical Analysis

The obtained information was subjected to statistical analysis using IBM SPSS Statistics Version 20.0 software (International Business Machines Corp., Armonk, NY, USA). The confidence interval (CI) was invariably calculated using confidence interval analysis (CIA) software (IBM SPSS Statistics Version 20.0). Before statistical analysis, the assumption of normality was tested using the Shapiro–Wilk test. Descriptive data were expressed as the mean ± standard deviation (SD), the median within the interquartile range (IQR), or the relative frequency with a 95% CI. The study applied specific tests for different types of analyzed data, including tests for comparing the mean values of a corresponding parameter across multiple data sets, such as an ANOVA and *t*-test, an correlation specific to quantitative and qualitative variables, including the Pearson chi-square (χ^2^) test. An association between tested variables was considered only when the calculated significance level (*p*-value) was less than the accepted level, *p* < 0.05 (with an accepted error rate of less than 5% of cases).

## 3. Results

The evaluation of the control group revealed the formation of a thick capsule, composed of mature connective tissue with orderly arranged collagen fibers. The fibrillar component, especially that of the collagen fibers, predominates over the cellular component (numerous collagen fibers and rare fibroblasts) (Figure 2).

The presence of inflammatory infiltrate was not observed. Fibroblasts and fibrocytes were in low numbers, and the capsule was uniform in both thickness and structure. The periprosthetic tissue zone exhibited a uniform and thick connective capsule surrounding the implanted material. This capsule consisted of mature, compact collagen fibers arranged linearly and interspersed with numerous newly formed blood capillaries, with the fibrillar component predominating the connective tissue. The surface in contact with the implanted material appeared to be smooth and regular.

In the rifampicin-treated group, the findings were akin to those observed in the control group. Moreover, proliferations of connective tissue were noted on the inner surface (in contact with the implanted material), resembling spicules (Figure 3).

The connective capsule after rifampicin treatment was well-represented, consisting of numerous linearly arranged collagen fibers interspersed with numerous newly formed blood vessels. The inflammatory infiltrate was absent.

In the third group treated with dexamethasone, a very thin capsule was identified (Figure 4a), with an immature appearance (predominance of fibroblasts over collagen fibers), and a disordered structure. A significant amount of inflammatory infiltrate was observed, represented in some areas by a giant-cell reaction (Figure 4b).

The connective capsule exhibited a heterogeneous structure alternating between areas of more compact connective tissue, composed of mature connective fibers arranged linearly, and areas where the capsule had reduced thickness, with immature and disordered connective fibers (either linear or in swirls). The connective capsule appeared to be lax, consisting of heterogeneous, undulating connective fibers arranged in a disorderly manner with numerous fibroblasts. It indicated connective tissue still in the process of organization. The very thin connective capsule was predominantly represented by fibroblasts and less by connective fibers. The contact surface with the implanted material was irregular, and a giant-cell reaction persisted. These aspects indicated a delayed tissue reaction from the host tissue. Collagenization after dexamethasone treatment appeared to be late compared to the other groups (significantly delayed collagenization). The tissue surrounding the implanted material exhibited an irregular connective capsule with uneven thickness around the implanted material. The circumference of the capsule displayed alternating thin and thick portions.

Treating the prosthesis with an autologous fat group was linked to the presence of a delicate connective capsule, characterized by mature collagen fibers arranged in a loosely organized manner (Figure 5a,b).

The capsule’s thickness was uniform throughout its circumference. At the capsule’s periphery, connective fibers were slightly interspersed with adipocytes, giving it a more lax structure with high elasticity (Figure 6a,b).

The inflammatory infiltrate was reduced. Newly formed blood vessels were present in the appropriate number. The interface with the implanted material appeared to be smooth.

The number of fibroblasts was low, indicating a completed collagenization process. The connective capsule after the local application of autologous fat was delicate, consisting of mature collagen fibers with a lax disposition. At the capsule’s periphery, connective fibers were slightly interspersed with adipocytes, giving it a more lax structure with high elasticity. The inflammatory infiltrate was absent. Newly formed blood vessels were present in the appropriate number. The contact surface with the implanted material was smooth (Figure 7).

The delicate conjunctival capsule is composed of mature collagen fibers with a lax disposition. The inflammatory infiltrate is absent or quantitatively reduced. The number of fibroblasts is low, indicating a completed collagenization process. Neoformation blood vessels are present in an appropriate number. The contact surface with the implanted material appears to be intact (Figure 8).

The delicate conjunctival capsule is composed of mature collagen fibers with a lax disposition. There is an absence of inflammatory infiltrate. The number of fibroblasts is low, indicating a completed collagenization process. Neoformation blood vessels are present in an appropriate number. The contact surface with the implanted material appears to be intact. The interface with the implanted material appears to be intact.

The evaluation of capsule parameters revealed a rich collagen fiber neogenesis in the control group and the rifampicin- and dexamethasone-treated groups, with a moderate number of collagen fibers in the autologous fat group. The angiogenesis was similar in all four groups, with a uniformly distributed increased intensity in the specimens. Leukocytic infiltrate was intensely represented in the dexamethasone-treated group, of low intensity in the control and rifampicin lavage groups, and absent in the autologous fat group. The intramuscular treatment with dexamethasone led to the formation of a capsule with uneven thickness compared to the other three study groups, where it was uniform. Areas of fibrosis included foci of granulomatous inflammation with multinucleated giant cells, reactive foreign body giant cells, macrophages loaded with hemosiderin pigment, lymphoplasmacytic inflammatory cells, and fibroblasts on the periphery. The autologous fat group stood out with a significantly smaller capsule thickness, which was weakly represented and featured a smooth surface.

In the second and third groups treated with dexamethasone and rifampicin, the capsule thickness was comparable, moderately represented, and appeared to be deformed, with the surface exhibiting spicules. The control group exhibited a smooth, uniform, and well-defined capsule (Table 1).

Pericapsular connective-muscle-adipose tissue presents multiple foci of granulomatous inflammation, with frequent multinucleated macrophages, incorporating synthetic textile material. Granulomas are associated with an abundant polymorphic inflammatory infiltrate rich in neutrophils and moderate fibrosis at the periphery.

### 3.1. Analysis of Periprosthetic Inflammation

In the current study, we observed both acute and chronic inflammatory reactions at the periprosthetic level, with variable intensities in all four groups. Perivascular inflammation, defined as the presence of inflammatory cells around blood vessels, was also scored on the same scale as mentioned above.

Acute inflammation was characterized by a high number of migrated neutrophils (Figure 9a) and the presence of immediate vascular changes (Figure 8b). Blood vessels were identified as endothelium-lined structures containing blood constituents (scattered red blood cells and white blood cells) and manually counted in the two most abundant areas at a magnification of ×40 (field diameter of 0.75 mm) (Figure 9b).

Chronic inflammation is predominantly characterized by the presence of monocytes, macrophages, and lymphocytes (Figure 10a) and is most commonly associated with toxicity or infection, while the foreign body reaction is most frequently composed of macrophages and foreign body giant cells. Persistent inflammatory stimuli lead to chronic inflammation (Figure 10b).

### 3.2. Statistical Study Results

The statistical analysis of the untreated prosthesis and the occurrence of inflammation revealed statistically significant correlations between acute and chronic inflammation, supporting the body’s reactivity to a foreign body (Table 2).

Rifampicin lavage has an anti-bactericidal role, thus allowing for the local control of infection, as identified by the absence of pericapsular acute inflammation and similar results for the percentage of chronic inflammation and active chronic inflammation as those obtained in the autologous fat group (Figure 11 and Figure 12).

Rifampicin lavage was statistically significantly associated with the absence of acute inflammation at the capsule level, emphasizing the importance of the bactericidal effect in limiting local infection and, consequently, inflammation (Table 3). The direct microscopic analysis of the capsules in the group where prostheses were lavaged with rifampicin did not detect bacterial colonies, and cultures confirmed the absence of microbial flora in this group.

Intramuscular injections of dexamethasone were associated with a relatively lower percentage of chronic inflammation and active chronic inflammation, but with a percentage of acute inflammation of 8.33%, much higher than the values obtained in the groups treated with autologous fat and rifampicin (Figure 13 and Figure 14).

The intramuscular administration facilitated the systemic modulatory effect on implant inflammation, with a different effectiveness time window depending on the type of medication and the route of administration in each group. Specific foreign body cells are the result of macrophage fusion when they are unable to phagocytize the foreign body and are associated with an anti-inflammatory and tissue remodeling environment, serving as a sign of chronic inflammation. No differences were found in the number of foreign body giant cells (FBGC) around the implant for any of the administered treatments, except for the control group, where an increase in FBGC was observed in the capsule around the implants compared to animals treated with dexamethasone. The effect obtained was evaluated at the time of prosthesis extraction together with the capsule, allowing for an estimation of the duration of this effect and the establishment of efficacy at various time points from silicone prosthesis insertion, a phenomenon valid for all four groups. Dexamethasone treatment significantly reduced the thickness of the capsule compared to the control group. However, the statistical analysis of the impact of dexamethasone on inflammation control did not identify statistically significant correlations between these parameters, without being able to establish whether the effect was dose-dependent, as all specimens received the same therapeutic regimen at a relatively close weight without significant changes in the dose of dexamethasone per kilogram of body weight (Table 4).

Treating the silicone prosthesis with autologous fat was linked with an absence of acute inflammation at the periprosthetic level. Chronic inflammation was predominant, accounting for 25% of cases, with active chronic inflammation being observed in 6.25% of instances (Figure 15 and Figure 16).

The statistical analysis of the impact of autologous fat on the occurrence of inflammation identified statistically significant correlations between acute inflammation and the application of autologous fat, suggesting its protective role in limiting inflammation and the rapid formation of a thin and fine capsule (Table 5). The distribution of fat in the examined specimens was uniform, and no significant differences influencing the results of this study were detected.

## 4. Discussion

The etiology of capsular contracture is still not fully understood, with silicone leaks, infection, foreign bodies, and surgical trauma identified as principal causes of bilateral capsular contracture. Unilateral capsule formation may primarily result from the formation of a hematoma or particles of devitalized tissue following trauma caused by an inadequate dissection or an excessive cauterization for hemostasis [18,30,31]. The silicone implant, considered a foreign body by the organism, triggers a periprosthetic capsular reaction, which is a natural response [32]. However, the reasons why some capsules contract remain partially unknown and are intensely studied [16,22]. In the present study, subjects in all four research groups developed peri-implant capsules, but the thickness of the capsules varied significantly depending on the locally or generally administered treatment. Ji Ung Park et al. demonstrated in an in vivo study that the implant initiates a foreign body reaction, leading to a cascade of inflammatory cell recruitment, fibroblast proliferation, and collagen synthesis, ultimately resulting in capsule formation. They also correlated the impact of capsule thickness with the appearance of capsular contracture, indicating a direct proportional relationship [33].

Prevention is the best way to treat capsular contractures, but the rate of their development has not decreased despite continuous efforts [8]. Capsular contracture remains a major complication of aesthetic and reconstructive breast augmentation. The incidence of capsular contracture has been the subject of multiple studies with variable results [34,35]. Two commonly accepted hypotheses are infection-related and hypertrophic scarring theories [36]. The infection-related theory questions the existence of a subclinical chronic infection located in close proximity to the implant envelope, manifesting as a microscopic biofilm. The biofilm is relatively inaccessible to cells and humoral immune function [37]. This process seems to be triggered by contamination.

The hypertrophic scarring theory suggests that the stimulation of myofibroblasts within the capsule leads to hypertrophic scar formation. These scars contract due to a series of phenomena similar to those seen in an inflammatory reaction [38,39,40].

Based on these hypotheses, the present study introduced two groups of subjects: one group underwent the biozonation of silicone implants with a rifampicin solution for infection prophylaxis, and the other group received intramuscular dexamethasone for its anti-inflammatory effect. In the current research, in the group treated with the biozonation of implants in a rifampicin solution before insertion into the created pocket, bacterial colonies were not detected on study slides, leading to the idea that locally placed antibiotics can prevent a potential infection but do not entirely reduce peri-implant capsule formation. In the present study, the use of rifampicin was chosen because there were studies that qualified it as a good bactericidal agent, especially for *Stapylococcus epidermidis*, as it is easy to use [41].

Reducing periprosthetic contracture has been reported using intraoperative irrigation with a 5% povidone-iodine solution [42]. Several authors recommend using systemic antibiotic administration (intravenous) intraoperatively and pocket irrigation with Bacitracin and/or Betadine. Concerns have been raised that agents used to disinfect breast pockets have harmful effects on wound healing [43,44].

The purely infectious etiology of capsular contracture is questioned by the reporting of positive bacterial cultures in peri-implant capsules in patients without capsular contracture. Other preventive and risk-reduction methods for capsular contracture are mentioned in the literature, including the use of steroids and immunomodulators to combat inflammation [45,46].

Local steroids have been used in the past to suppress capsular contracture [46]. They were introduced directly into the pocket or into an inflatable or double-lumen implant [47,48]. Three-quarters of a series of patients treated with a technique recommended in 1976 later required the removal or replacement of implants due to severe ptosis or soft tissue atrophy, and 13% of the same series developed capsular contracture despite steroid administration. Mladick also suggests that the use of intraluminal steroids or pocket irrigation is not recommended [49].

Efforts have been made to control inflammation after breast augmentation procedures to prevent capsular contracture. The application of anti-inflammatory strategies has been well reviewed in many studies. Study results support the idea that limiting inflammation during the wound healing phase inhibits fibrosis [24,25,50].

Mojallal and colleagues demonstrated collagen fiber neosynthesis and subsequent dermal thickening in an animal model after the subdermal injection of human adipose tissue [50]. Recent studies suggest that adipose tissue contains a cellular fraction (stromal cells and/or stem cells derived from adipocytes) that contributes to improving wound healing, tissue repair, and extracellular matrix remodeling [31,51]. After centrifuging the collected fat, a separation is obtained into three portions of the content: oil, fat, and fluid with destroyed cells—only the fat portion will be used. Papadopoulos et al. [52] found that autologous adipose tissue transfer could alleviate the pain caused by capsular contracture and reduce the degree of contracture when it occurs. The treatment involves several stages of fat tissue injection around the implant, requiring a longer time interval.

The authors attributed the pain relief to the differentiation and softening of tissues, leading to a reduced compression on the nerves [52].

The limitation of the study is the relatively short follow-up period for the subjects.

The six-week follow-up period recorded the follow-up interval between 30 and 90 days reported in the literature [53]. The comparative histological evaluation method using autologous fat shows the effectiveness of its use, indicating the absence of acute inflammation and the formation of a thin and fine capsule.

## 5. Conclusions

Untreated periprosthetic material induces a significant foreign body reaction, characterized by increased amounts of inflammatory infiltrates. The formation of a consistent, uniform connective capsule, composed of neoformed mature collagen fibers, takes place, all these favoring the occurrence of capsular contracture

The use of autologous fat limits the “non-self” reaction. This will favor the formation of a periprosthetic capsule made of mature collagen fibers, having a loose and uniform disposition throughout the circumference. This interspersed structure with adipocytes will give elasticity to the capsule.

The foreign body reaction, compared to other studied groups, was reduced, defining the protective role of autologous fat in the occurrence of capsular contracture, minimizing the risk of scar formation. The current study may serve as the basis for future therapeutic protocols in the use of breast implants, with the use of autologous fat being a safe method in terms of organism reactivity and associated with minimal costs.

## Figures and Tables

**Figure 1 diagnostics-14-00661-f001:**
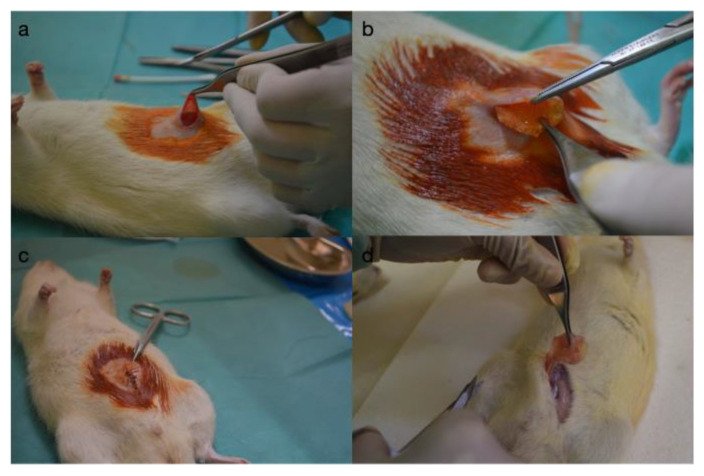
Intraoperative view: (**a**) creating the pocket for the implant, (**b**) the insertion of the implant, (**c**) the closure of the operative wound, and (**d**) the extracting the implant and the capsule.

**Figure 2 diagnostics-14-00661-f002:**
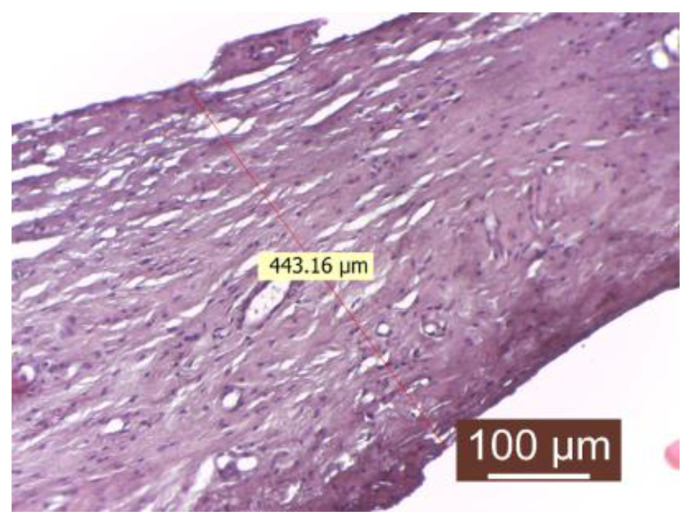
Peripheral tissue of the implanted material in the control group (HEx40). The peripheral tissue zone of the implant is represented by a substantial (thick) conjunctival capsule.

**Figure 3 diagnostics-14-00661-f003:**
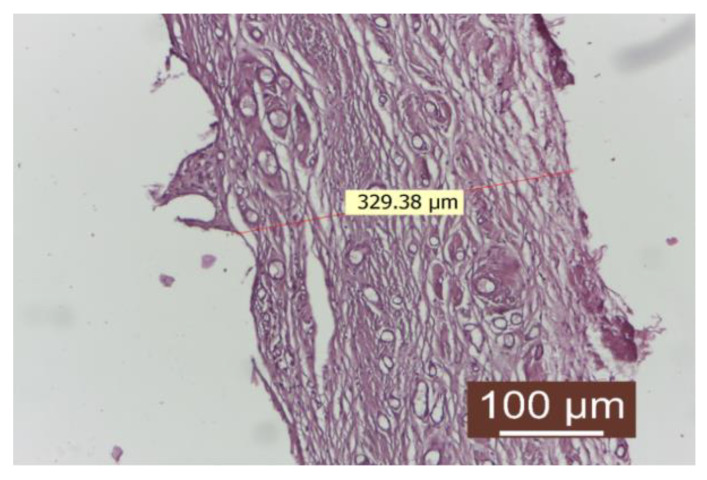
Peripheral tissue of the implanted material in the rifampicin-treated group (HEx10). There is a well-represented conjunctival capsule, composed of numerous linearly arranged collagen fibers interspersed with numerous neovascular blood vessels. The inflammatory infiltrate is absent.

**Figure 4 diagnostics-14-00661-f004:**
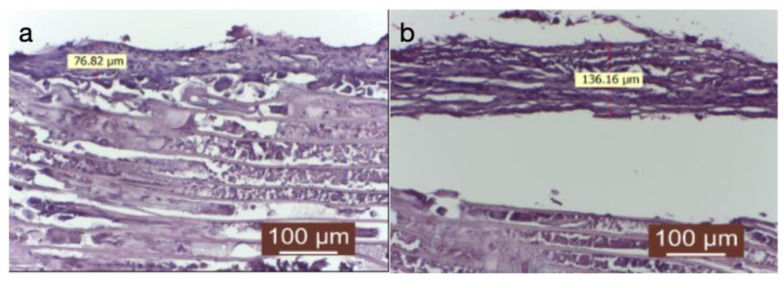
(**a**) Implanted material and its peripheral tissue in the group treated with dexamethasone (HEx40). Ther is a very thin conjunctival capsule, mainly composed of fibroblasts and fewer connective fibers. (**b**) The contact surface with the implanted material is irregular, and a giant-cell reaction still persists. These aspects indicate a delayed tissue reaction from the host tissue.

**Figure 5 diagnostics-14-00661-f005:**
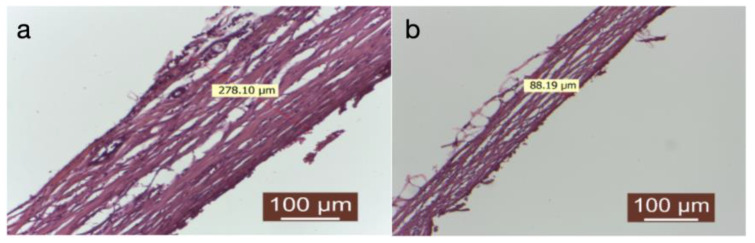
(**a**,**b**) Group treated with autologous fat. The peripheral tissue surrounding the implanted material was assessed, including the measurement of the capsule (HEx40).

**Figure 6 diagnostics-14-00661-f006:**
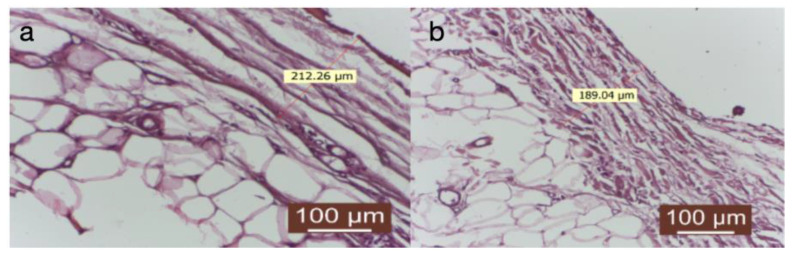
(**a**,**b**) Group treated with autologous fat: the thickness of the capsule is uniform throughout its circumference. At the periphery of the capsule, the connective fibers are slightly interspersed with adipocytes, giving it a looser structure, with greater elasticity (HEx40).

**Figure 7 diagnostics-14-00661-f007:**
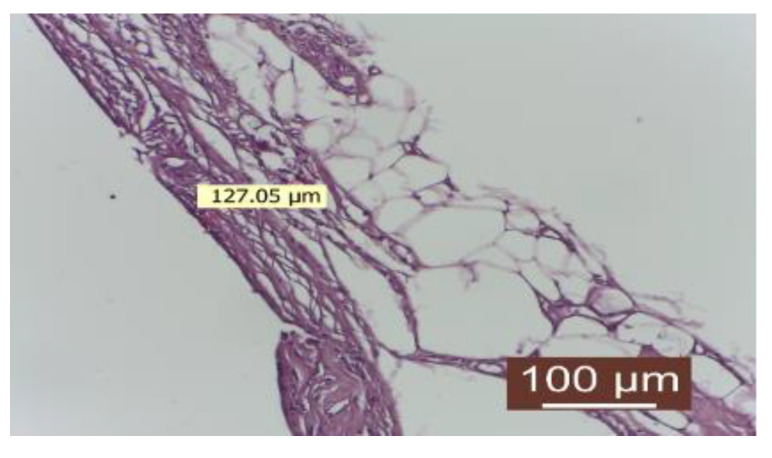
Group treated with autologous fat: there is a delicate conjunctive capsule, consisting of mature collagen fibers with a loose disposition. At the periphery of the capsule, connective fibers are slightly interspersed with adipocytes, giving it a looser structure with high elasticity. The inflammatory infiltrate is absent (HEx40).

**Figure 8 diagnostics-14-00661-f008:**
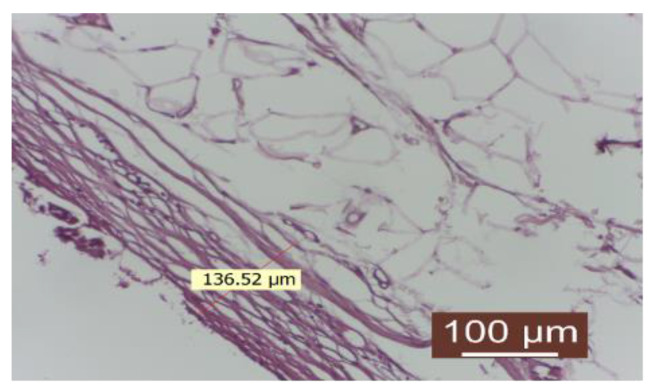
Group treated with autologous fat: the inflammatory infiltrate is absent. The number of fibroblasts is reduced, revealing a completed collagenization process. There is an appropriate number of neoformation blood vessels. The contact surface with the implant material is smooth (HEx40).

**Figure 9 diagnostics-14-00661-f009:**
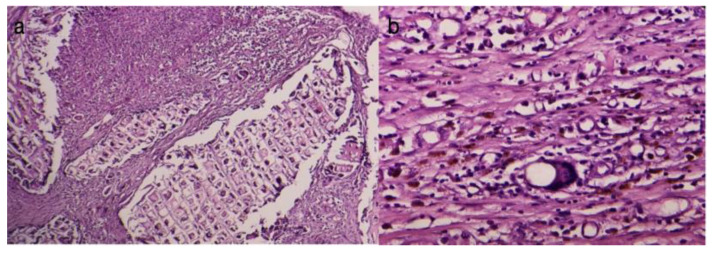
(**a**) Acute periprosthetic inflammation (HEx20), (**b**) Acute pericapsular inflammation (HEx40).

**Figure 10 diagnostics-14-00661-f010:**
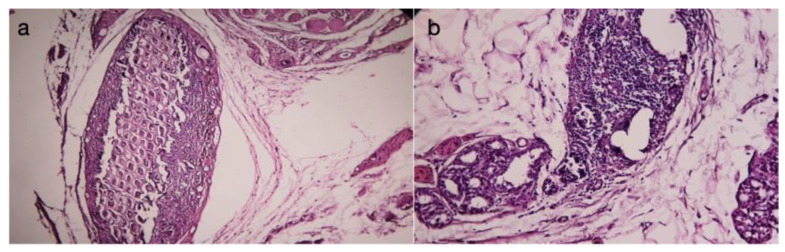
(**a**) Chronic periprosthetic inflammation (HEx40), (**b**) Chronic inflammation (HEx40).

**Figure 11 diagnostics-14-00661-f011:**
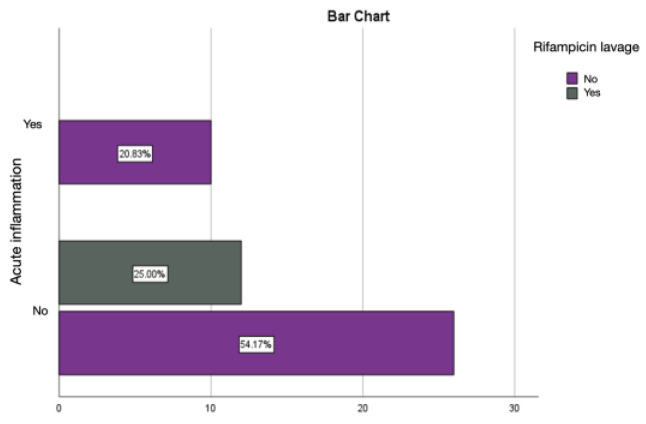
Assessment of acute inflammation in the group with prostheses immersed in the rifampicin solution.

**Figure 12 diagnostics-14-00661-f012:**
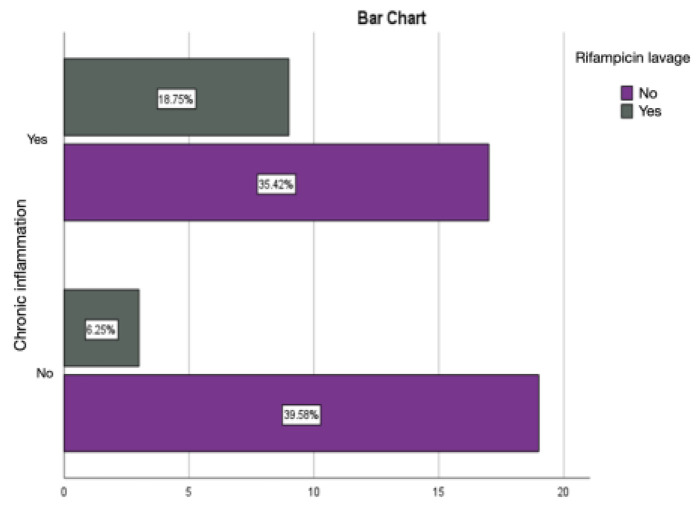
Assessment of chronic inflammation in the group with prostheses immersed in the rifampicin solution.

**Figure 13 diagnostics-14-00661-f013:**
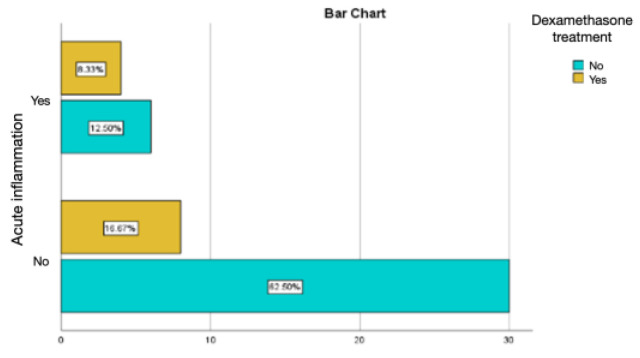
Assessment of acute inflammation in the group with the intramuscular administration of dexamethasone.

**Figure 14 diagnostics-14-00661-f014:**
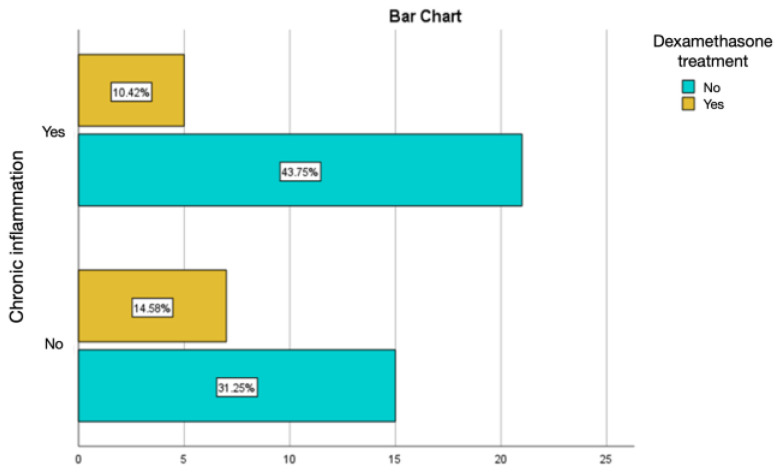
Assessment of chronic inflammation in the group with the intramuscular administration of dexamethasone.

**Figure 15 diagnostics-14-00661-f015:**
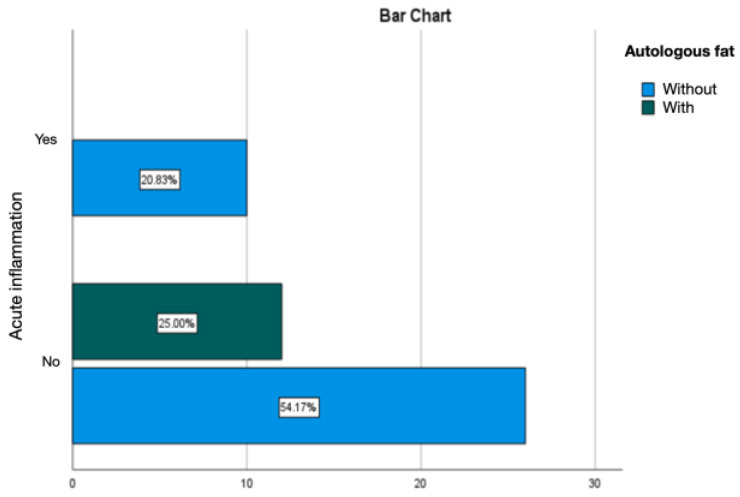
Evaluation of acute inflammation in the group treated with autologous fat.

**Figure 16 diagnostics-14-00661-f016:**
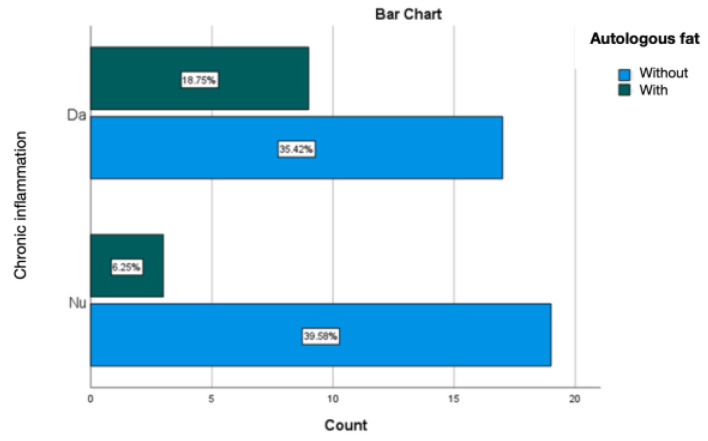
Evaluation of chronic inflammation in the group treated with autologous fat.

**Table 1 diagnostics-14-00661-t001:** Morphopathological characteristics of the periprosthetic capsule.

Evaluation Criteria for the Capsule	Control	Rifampicin	Dexamethasone	Autologous Fat
Fibrillar neogenesis (collagen fibers)	+++	+++	+++	++
Neogenesis of newly formed blood vessels	+++	+++	+++	+++
Leukocytic infiltrate	+	+	+++	-
Thickness of the connective capsule	+++ uniform	++ uniform	++ non-uniform	+ uniform
Uniformity of the capsule thickness	smooth	deformed (spicules)	deformed (spicules)	smooth

+—low, ++—moderate, +++—high.

**Table 2 diagnostics-14-00661-t002:** Statistical correlations between inflammation and prosthesis presence in the control group.

		Acute Inflammation	Chronic Inflammation	Active Chronic Inflammation	Without Prosthesis Treatment
Acute inflammation	Pearson Correlation	1	−0.455 **	−0.103	0.415 **
	Sig. (2-tailed)		0.001	0.484	0.003
	N	48	48	48	48
Chronic inflammation	Pearson Correlation	−0.455 **	1	−0.422 **	−0.338 *
	Sig. (2-tailed)	0.001		0.003	0.019
	N	48	48	48	48
Active chronic inflammation	Pearson Correlation	−0.103	−0.422 **	1	0.159
	Sig. (2-tailed)	0.484	0.003		0.281
	N	48	48	48	48
Without prosthesis treatment	Pearson Correlation	0.415 **	−0.338 *	0.159	1
	Sig. (2-tailed)	0.003	0.019	0.281	
	N	48	48	48	48

* Correlation is significant at the 0.05 level (2-tailed). ** Correlation is significant at the 0.01 level (two-tailed).

**Table 3 diagnostics-14-00661-t003:** Correlations between inflammation and the rifampicin lavage of the prosthesis.

		Acute Inflammation	Chronic Inflammation	Active Chronic Inflammation	Rifampicin Lavage
Acute inflammation	Pearson Correlation	1	−0.455 **	−0.103	−0.296 *
	Sig. (2-tailed)		0.001	0.484	0.041
	N	48	48	48	48
Chronic inflammation	Pearson Correlation	−0.455 **	1	−0.422 **	0.241
	Sig. (2-tailed)	0.001		0.003	0.098
	N	48	48	48	48
Active chronic inflammation	Pearson Correlation	−0.103	−0.422 **	1	−0.053
	Sig. (2-tailed)	0.484	0.003		0.721
	N	48	48	48	48
Rifampicin lavage	Pearson Correlation	−0.296 *	0.241	−0.053	1
	Sig. (2-tailed)	0.041	0.098	0.721	
	N	48	48	48	48

* Correlation is significant at the 0.05 level (2-tailed). ** Correlation is significant at the 0.01 level (2-tailed).

**Table 4 diagnostics-14-00661-t004:** Correlations of dexamethasone treatment and periprosthetic inflammation.

		Acute Inflammation	Chronic Inflammation	Active Chronic Inflammation	Dexamethasone Treatment
Acute inflammation	Pearson Correlation	1	−0.455 **	−0.103	0.178
	Sig. (2-tailed)		0.001	0.484	0.227
	N	48	48	48	48
Chronic inflammation	Pearson Correlation	−0.455 **	1	−0.422 **	−0.145
	Sig. (2-tailed)	0.001		0.003	0.326
	N	48	48	48	48
Active chronic inflammation	Pearson Correlation	−0.103	−0.422 **	1	−0.053
	Sig. (2-tailed)	0.484	0.003		0.721
	N	48	48	48	48
Dexamethasone treatment	Pearson Correlation	0.178	−0.145	−0.053	1
	Sig. (2-tailed)	0.227	0.326	0.721	
	N	48	48	48	48

** Correlation is significant at the 0.01 level (2-tailed).

**Table 5 diagnostics-14-00661-t005:** Statistical correlations between inflammation and the autologous fat treatment.

		Acute Inflammation	Chronic Inflammation	Active Chronic Inflammation	Autologous Fat
Acute inflammation	Pearson Correlation	1	−0.455 **	−0.103	−0.296 *
	Sig. (2-tailed)		0.001	0.484	0.041
	N	48	48	48	48
Chronic inflammation	Pearson Correlation	−0.455 **	1	−0.422 **	0.241
	Sig. (2-tailed)	0.001		0.003	0.098
	N	48	48	48	48
Active chronic inflammation	Pearson Correlation	−0.103	−0.422 **	1	−0.053
	Sig. (2-tailed)	0.484	0.003		0.721
	N	48	48	48	48
Autologous fat	Pearson Correlation	−0.296 *	0.241	−0.053	1
	Sig. (2-tailed)	0.041	0.098	0.721	
	N	48	48	48	48

* Correlation is significant at the 0.05 level (2-tailed). ** Correlation is significant at the 0.01 level (2-tailed).

## Data Availability

Data is contained within the article.

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
