# Peer review of "A Histological Evaluation of the Efficiency of Using Periprosthetic Autologous Fat to Prevent Capsular Contracture Compared to Other Known Methods—An Experimental Study"

_diagnostics, 2024, doi:10.3390/diagnostics14060661_

Round 1

Reviewer 1 Report

Comments and Suggestions for Authors

The study gives us clear histological information about the efficiency of using periprosthetic autologous fat to prevent capsular contracture.

The reviewer thinks it is worth being published. The reviewer recommends minor revision as follows.

Please display the images of surgery performed on the animals.

Line 137-138; “The collected periprosthetic capsule samples were fixed in 10% buffered formalin. After 24 hours, they were embedded in paraffin.” might be some mitake.

Line 143; What is “astectule ratio” ? Typo?

Author Response

Dear Reviewer 1,

Thank you for reviewing the manuscript and for the recommendations.

According to your recommendations, I made the following additions:

  • I have displayed the images of surgery performed on the animals – line 121 (Figure 1)
  • At lines 137-138 - I corrected: ,,after 48 h,,
  • Line 143 : I corrected: ,,fibroblast/collagen fiber/astectule ratio and collagen fiber appearance and density),, - now it’s line 149.

Thank you very much!

Best regards,

Mihaela Pertea MD PhD, Assoc Prof

Reviewer 2 Report

Comments and Suggestions for Authors

Thank you for having me to review this manuscript. I suggested some points to improve reading through this research.

Introduction

The author mentioned advantages and disadvantages of antibiotics and steroid insertion with implants, however the author didn't mention about autologous fat insertion.

Please describe about autologous fat insertion more detail in introduction.

Results

I couldn't understand what is important in this result.

The author should show the data separately in three groups and avoid to change the paragraph so much.

I couldn't understand the sentence is indicating which of three groups.

What is the difference of fig 4, 5, 6 and 7?

I couldn't find the difference of them from explanation in each figure.

Comments on the Quality of English Language

Nothing to say.

Author Response

Dear Reviewer 2,

Thank you for reviewing the manuscript and for the recommendations.

According to your recommendations, I made the following additions:

  • I completed the introduction with mentions about autologous fat insertion: lines 80-87, with the introduction of a new bibliographic reference [26]
  • the groups studied were described in the lines 123-130
  • At the results I mentioned the groups: ,,control group,, (line 173), ,,Rifampicin-treated group,, (line 188), ,,group treated with Dexamethasone,, (199), ,,Treating the prosthesis with autologous fat group,, (223)
  • I reviewed the explanations of the figures 5,6,7

Thank you very much!

Best regards,

Mihaela Pertea MD PhD

Reviewer 3 Report

Comments and Suggestions for Authors

Your study importantly demonstrates that the use of autologous fat decreases capsule thickness around the implant.

Please discuss:

1. Why was the capsule harvested at 6 weeks for examination after implant placement?

2.What portion of the centrifuged fat was injected around the implant?

3.Why was rifampicin chosen as the antibiotic?

Please correct:

1.On page 11 Line 340 there is a typing error: “groups treated with autologou/s fat and Rifampicin (Figures. 12, 13).”

Author Response

Dear Reviewer 3,

Thank you for reviewing the manuscript and for the recommendations.

According to your recommendations, I made the following additions:

  • I discussed: Why was the capsule harvested at 6 weeks for examination after implant placement (lines 489-490) with the introduction of a new bibliographic reference (54)

What portion of the centrifuged fat was injected around the implant – lines   480-482

Why was rifampicin chosen as the antibiotic – lines 451-454) with the introduction of a new bibliographic reference (43)

  • I corrected line 340 (now 354) - : “groups treated with autologou/s fat and Rifampicin (Figures. 12, 13).”

Best regards,

Mihaela Pertea MD PhD, Assoc Prof
